# Axial light loss of monocytes as a readily available prognostic biomarker in patients with suspected infection at the emergency department

**Titus A. P. de Hond**[1]*, **Wout J. Hamelink**[1], **Mark C. H. de Groot**[2], **Imo E. Hoefer**[2], **Jan Jelrik Oosterheert**[3], **Saskia Haitjema**[2], **Karin A. H. Kaasjager**[1]

1 Department of Internal Medicine and Acute Medicine, University Medical Centre Utrecht, Utrecht University, Utrecht, The Netherlands, 2 Central Diagnostic Laboratory, Division Laboratory, Pharmacy and Biomedical Genetics, University Medical Centre Utrecht, Utrecht University, Utrecht, The Netherlands, 3 Department of Internal Medicine and Infectious Diseases, University Medical Centre Utrecht, Utrecht University, Utrecht, The Netherlands

* t.a.p.dehond@umcutrecht.nl

**Data Availability Statement:** Data will not be publicly shared due to ethical and privacy restrictions. The dataset contains sensitive,

## Abstract

### Objectives

To evaluate the prognostic value of the coefficient of variance of axial light loss of monocytes (cv-ALL of monocytes) for adverse clinical outcomes in patients suspected of infection in the emergency department (ED).

### Methods

We performed an observational, retrospective monocenter study including all medical patients ≥18 years admitted to the ED between September 2016 and June 2019 with suspected infection. Adverse clinical outcomes included 30-day mortality and ICU/MCU admission <3 days after presentation. We determined the additional value of monocyte cv-ALL and compared to frequently used clinical prediction scores (SIRS, qSOFA, MEWS). Next, we developed a clinical model with routinely available parameters at the ED, including cv-ALL of monocytes.

### Results

A total of 3526 of patients were included. The OR for cv-ALL of monocytes alone was 2.21 (1.98–2.47) for 30-day mortality and 2.07 (1.86–2.29) for ICU/MCU admission <3 days after ED presentation. When cv-ALL of monocytes was combined with a clinical score, the prognostic accuracy increased significantly for all tested scores (SIRS, qSOFA, MEWS). The maximum AUC for a model with routinely available parameters at the ED was 0.81 to predict 30-day mortality and 0.81 for ICU/MCU admission.

### Conclusions

Cv-ALL of monocytes is a readily available biomarker that is useful as prognostic marker to predict 30-day mortality. Furthermore, it can be used to improve routine prediction of adverse clinical outcomes at the ED.

personal information and will therefore be only available upon reasonable request to the ethics committee of the UMC Utrecht (metc@umcutrecht. nl).

**Funding:** The author(s) received no specific funding for this work.

**Competing interests:** The authors have declared that no competing interests exist.

## Clinical trial registration

Registered in the Dutch Trial Register (NTR) und number 6916.

## 1. Introduction

Sepsis is defined as a life-threatening organ dysfunction caused by a dysregulated host response to infection [1]. It is a clinical syndrome that is known to have high morbidity and mortality rates [2, 3]. Unfortunately, no accurate diagnostic tools are available for early recognition of sepsis [4–7]. Clinical prediction scores (e.g. SIRS, (q)SOFA or Modified Early Warning Score (MEWS)) have been developed for recognition of severely ill patients in an Emergency Department (ED) setting [1, 4, 8–10] but poorly predict adverse clinical outcomes [11–14]. In addition, these scores consist of different patient characteristics that need to be collected manually and processed in the electronic health record (EHR) system to perform optimally. Moreover, scores such as the MEWS were not specifically developed in the context of sepsis, but rather to predict outcome in a range of critically ill patients [11, 15]. Therefore, there is a continuous need for easy, accurate and cheap accessible biomarkers for prediction of adverse clinical outcomes in sepsis patients, especially early in the course of the disease. Recently, numerous biomarkers have been identified, but these are mostly costly and therefore complicate the chase for value-based healthcare.

Leukocytes play a key role in the inflammatory host response to infection [16, 17] and are therefore used as biomarker for the disease [18]. Nevertheless, leukocytes are nonspecific and consist of multiple cell subsets [17, 19] that may be more specific and more accurate biomarkers for sepsis [17]. Specifically monocytes, as part of the innate immune system, play a crucial role in the very early stage of sepsis [20, 21]. In early stages of sepsis monocytes are activated and undergo morphological changes [21, 22] that may be useful for early identification of the disease. Recently, Monocyte Distribution Width (MDW) was suggested as an early sepsis indicator [22–27]. A downside to MDW as a biomarker is the requirement of a specific costly analyzer [22]. Another approach to calculate the variety in monocyte size uses the flow cytometry principle within existing hematology analyzers to assess leukocyte subsets. The axial light loss (ALL) or 'shadow' that is routinely obtained as a cell passes the laser light inside the machine during such a measurement can be seen as a proxy of cell size. In raw hematology data ALLs are available as means with accompanying coefficients for different subsets of leukocytes. Coefficient of variance of axial light loss of monocytes (cv-ALL of monocytes) can be seen as a way to express variety in monocytic volumetric size, and is thereby very much comparable to MDW.

Therefore, we used readily available cv-ALL of monocytes to study monocyte characteristics as a biomarker for clinical outcome. We hypothesized that cv-ALL of monocytes is a valuable biomarker to predict clinical adverse outcomes in patients that are suspected of an infection at the ED.

## 2. Methods

### 2.1 Study design

We performed an observational retrospective cohort study on data from the SPACE-cohort (SePsis in the Acutely ill patients in the Emergency department) [28] that was collected between September 2016 and September 2019. The SPACE-cohort includes all patients with suspected infection presenting in the ED of the University Medical Centre Utrecht (UMCU)

that fulfill the following 2 inclusion criteria: ≥18 years, and presenting for the internal medicine department or one of its subspecialities. No other in- or exclusion criteria are used.

All patients in the SPACE-cohort were assessed for the presence of sepsis. If sepsis was suspected a sepsis care pathway was initiated, resulting in protocolized care. Non-septic patients received standard of care treatment according to their clinical situation. The SPACE-cohort was reviewed and approved by the Medical Ethical Committee of the UMCU under number 16/594 and registered in the Dutch Trial Register (NTR) under number 6916.

## 2.2 Data collection

The treating physician at the ED is always asked by the EHR system whether the patient is suspected of an infection and whether it could be sepsis in our center. If both questions are answered positively, the system automatically calculates the SIRS and qSOFA scores using the first set of vital parameters obtained during the ED visit. If no such parameters are available in the system, lacking parameters can be added manually. When at least one of the scores is abnormal, the physician is alerted by a screen warning message. These patients were automatically included in the SPACE cohort.

As secondary quality check for completeness of the SPACE cohort, independent trained clinicians screened all patient records of ED visits for the suspicion of infection and/or sepsis if registration via the clinical pathway was absent. If an infectious cause was mentioned in the differential diagnosis, patients received antibiotics, or bacterial cultures were taken these patients were considered to be suspected of infection and were also included in the SPACE-cohort.

For all included patients, data concerning demographics, vital parameters, antibiotics, comorbidities, and outcome was collected manually and supplemented with automated queries for laboratory variables to calculate cv-ALL of monocytes. Data on vital parameters included all parameters to calculate clinical prediction scores (SIRS, qSOFA, MEWS) and follow-up data on morbidity and mortality included microbiological diagnostics, chosen treatment, hospitalization, and length of stay). Charlson Comorbidity Index (CCI) was used for the collection of comorbidities [29].

## 2.3 Biochemical parameters

Standardized blood draw was performed at the ED including a complete blood count (CBC). In the UMC Utrecht, raw data including the full optical parameters of each measured individual blood cell is extracted automatically from the hematological analyzer (Abbott CELL-DYN Sapphire) and stored into the Utrecht Patient Oriented Database (UPOD). The structure and content of UPOD have been described in more detail elsewhere [30]. From this raw data we extracted the cv-ALL of monocytes.

## 2.4 Outcomes

The primary outcome of this study was 30-day all-cause mortality [31, 32] and secondary endpoints were Medium Care Unit (MCU) or Intensive Care Unit (ICU) admission <3 days after ED presentation. For the secondary outcome, all patients with an ICU-restrictive policy were excluded.

## 2.5 Statistical analyses

Normally distributed continuous data are expressed as a mean with standard deviation (SD). Non-parametric data are shown as median and interquartile range (IQR). Student's t test was used to compare normally distributed continuous parameters, while a Mann Whitney U test was used for non-parametric continuous variables. Categorical variables were compared using

Chi-Square or Fisher's exact test, depending on variable size. We used a predictive mean matching multiple imputation approach for missing values. All included vital, laboratory and outcome parameters that were used in our analyses were used. Concerning data points on laboratory variables, hospital admission, and clinical course, all used parameters had missing data <1%. This was also the case for all used vital parameters, except for respiratory rate (missing 25.8%). No data on demographics were missing.

We studied the association between cv-ALL of monocytes and outcomes using binary logistic regression models. First, we compared the predictive value of cv-ALL of monocytes to frequently used clinical prediction scores (SIRS, qSOFA, MEWS). The optimal cut-off point for cv-ALL of monocytes was calculated via Youden's statistic. Next, we tested the additional value of cv-ALL of monocytes on top of these scores. We assessed additional value using likelihood ratio tests. Finally, using stepwise regression via backward selection, we combined all individual parameters from the clinical scores, patient characteristics, and cv-ALL of monocytes to come up with a clinical model with easily accessible parameters. Prognostic accuracy was evaluated by receiver operating characteristic (ROC) curve analyses and reported as area under the curve (AUC) with 95% confidence intervals. Calibration curves were constructed with R Statistical Software, version 4.0.3. The following packages were used: haven, tidyverse, and rms. IBM SPSS Statistics version 26.0 was used for all other analyses and p-values below 0.05 were considered statistically significant.

## 3. Results

### 3.1 Patient characteristics

A total of 3526 patients were enrolled. Table 1 shows the baseline characteristics of the cohort. Patients were on average 61.0 years old (53.4% male). Median of cv-ALL of monocytes in the whole cohort was 0.077 (IQR 0.070–0.088). Cv-ALL of monocytes was associated with disease severity (S1 and S2 Figs). The magnitude of cv-ALL of monocytes increases in sicker patients (S1 Fig). The percentage of patients with a high cv-ALL of monocytes measurement increased if SIRS, qSOFA or MEWS get higher (S2 Fig). Additionally, we hypothesized that cv-ALL of monocytes might differ between immunocompromised and non-immunocompromised patients and indeed, there was a significant difference between these two groups (S3 Fig).

### 3.2 Primary outcome

The overall 30-day mortality was 6.3% (222/3526 patients). The median cv-ALL of monocytes in survivors and non-survivors was 0.077 (IQR 0.070–0.088) vs 0.084 (IQR 0.073–0.098), p <0.001. The optimal cut-off point to predict 30-day mortality was 0.085. This dichotomization resulted in an OR of 2.21 (95% CI 1.98–2.47, Table 2). Based on the likelihood ratio tests, cv-ALL of monocytes had an additional predictive value to every clinical prediction score (Fig 1A, Table 3). The best multivariable logistic regression model contained cv-ALL of monocytes, age, sex, CCI, respiratory rate, systolic blood pressure, Glasgow Coma Scale, heart rate, white blood cell count and body temperature as independent factors associated with 30-day mortality. The corresponding ROC curves are shown in Fig 2A, with an AUC for this model of 0.81. Calibration curve of the optimal model is shown in S4 Fig with $R^2$ of 0.209 and Brier score of 0.054.

### 3.3 Secondary outcome

Within 3 days after ED visit, 8.6% (303/3526 patients) were admitted to MCU and/or ICU. The optimal cut-off point for cv-ALL of monocytes was 0.088, corresponding with an OR of

**Table 1. Baseline table of the SPACE population.**

| | Total (n = 3526) | Survivors (n = 3304) | Non-survivors (n = 222) | P-value |
|---|---|---|---|---|
| **Demographic** | | | | |
| Age–yr–median (IQR) | 61.0 (48.0–70.0) | 61.0 (46.0–70.0) | 68.0 (59.0–75.0) | <0.001 |
| Sex, male (%) | 1884 (53.4) | 1735 (52.5) | 149 (67.1) | <0.001 |
| CCI (≥ 5) (%) | 1683 (47.7) | 1503 (45.5) | 180 (81.1) | <0.001 |
| **Specialties** | | | | <0.001 |
| Internal medicine (%) | 1088 (30.9) | 1029 (31.1) | 59 (26.6) | |
| Nephrology (%) | 571 (16.2) | 559 (16.9) | 12 (5.4) | |
| Oncology (%) | 615 (17.4) | 536 (16.2) | 79 (35.6) | |
| Hematology (%) | 574 (16.3) | 531 (16.1) | 43 (19.4) | |
| Rheumatology (%) | 207 (5.9) | 200 (6.1) | 7 (3.2) | |
| Endocrinology (%) | 124 (3.5) | 123 (3.7) | 1 (0.5) | |
| Infectious diseases (%) | 74 (2.1) | 73 (2.2) | 1 (0.5) | |
| Other (%) | 273 (7.7) | 253 (7.7) | 20 (9.0) | |
| **Clinical scores** | | | | |
| SIRS score ≥2 (%) | 2194 (62.2) | 2025 (61.3) | 169 (76.1) | <0.001 |
| qSOFA score ≥2 (%) | 195 (5.5) | 154 (4.7) | 41 (18.5) | <0.001 |
| MEWS ≥5 (%) | 498 (14.1) | 425 (12.9) | 73 (32.9) | <0.001 |
| **Timing of antibiotics** | | | | <0.001 |
| No antibiotics | 2136 (60.6) | 2051 (62.1) | 85 (38.3) | |
| <1 hour | 148 (4.2) | 129 (3.9) | 19 (8.6) | |
| 1–3 hours | 568 (16.1) | 515 (15.6) | 53 (23.9) | |
| >3 hours | 674 (19.1) | 609 (18.4) | 65 (29.3) | |
| **Clinical course** | | | | |
| Hospital admission (%) | 2307 (65.4) | 2113 (64.0) | 194 (87.4) | <0.001 |
| Length of stay–days–median (IQR) | 4.7 (2.7–8.7) | 4.7 (2.7–8.5) | 5.6 (2.5–12.7) | 0.069 |
| **Cv-ALL of monocytes** | | | | |
| Median cv-ALL of monocytes (IQR) | 0.077 (0.070–0.088) | 0.077 (0.070–0.088) | 0.084 (0.073–0.098) | <0.001 |

CCI, Charlson Comorbidity Index; cv-ALL, coefficient of variance of axial light loss

**Table 2. Univariate logistic regression for 30-day mortality.**

| Predictor | OR (95% CI) | p-Value |
|---|---|---|
| **30-day mortality** | | |
| cv-ALL (≥0.085) | 2.21 (1.98–2.47) | <0.001 |
| SIRS (≥2) | 1.88 (1.65–2.14) | <0.001 |
| qSOFA (≥2) | 4.57 (3.932–5.32) | <0.001 |
| MEWS (≥5) | 3.06 (2.72–3.46) | <0.001 |
| **ICU/MCU admission <3 days** | | |
| cv-ALL (≥0.088) | 2.07 (1.86–2.29) | <0.001 |
| SIRS (≥2) | 3.32 (2.89–3.80) | <0.001 |
| qSOFA (≥2) | 10.10 (8.82–11.56) | <0.001 |
| MEWS (≥5) | 5.60 (5.04–6.22) | <0.001 |

cv-ALL, Coefficient of Variance Axial Light Loss; qSOFA, quick Sequential Organ Failure Assessment; OR, odds ratio; CI, confidence interval.

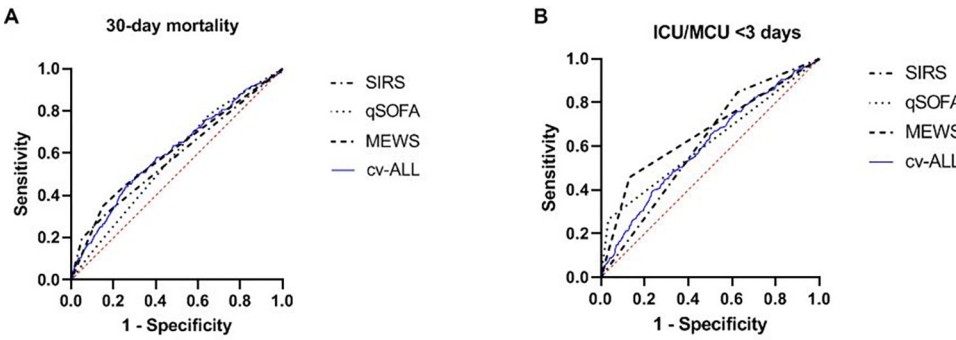

**Fig 1.** ROC curves to predict 30-day mortality (A) and ICU/MCU admission <3 days after ED presentation (B). The AUC of cv-ALL of monocytes to predict 30-day mortality (AUC = 0.61) was higher than the AUC of the clinical scores SIRS (AUC = 0.57) and qSOFA (AUC = 0.57), and comparable to MEWS (AUC = 0.61). For the prediction of ICU/MCU admission the AUC of cv-ALL of monocytes (AUC = 0.60) was slightly lower than the AUC of the clinical scores: SIRS (AUC = 0.61), qSOFA (AUC = 0.62), and MEWS (AUC = 0.66).

2.07 (95% CI 1.86–2.29, Table 2). The AUC for cv-ALL of monocytes was lower than for SIRS, qSOFA, and MEWS (AUC 0.60 vs 0.61 vs 0.62 vs 0.66 respectively, Fig 1B, Table 3). Again, cv-ALL of monocytes added significantly to the model performance of each clinical score (Table 3). In multivariable regression analysis cv-ALL of monocytes, age, sex, CCI, respiratory rate, systolic blood pressure, Glasgow Coma Scale, heart rate, white blood count and body temperature were independent predictors for ICU/MCU admission <3 days after ED presentation. The maximum AUC for this model was 0.81 (Fig 2B), with calibration curve shown in S4 Fig ($R^2$ = 0.220; Brier score = 0.070). Unlike for our primary outcome, CCI was negatively correlated with ICU/MCU admission, meaning a higher CCI was associated with a lower chance of being admitted to the ICU/MCU <3 days.

**Table 3. AUC of clinical prediction scores with(out) cv-ALL of monocytes.**

| Predictor | AUC | LRT |
|---|---|---|
| **30-day mortality** | | |
| cv-ALL | 0.61 | - |
| SIRS (≥2) | 0.57 | |
| SIRS + cv-ALL | 0.62 | <0.001 |
| qSOFA (≥2) | 0.57 | |
| qSOFA + cv-ALL | 0.64 | <0.001 |
| MEWS (≥5) | 0.60 | |
| MEWS + cv-ALL | 0.65 | <0.001 |
| **ICU/MCU admission <3 days** | | |
| cv-ALL | 0.60 | - |
| SIRS (≥2) | 0.61 | |
| SIRS + cv-ALL | 0.66 | <0.001 |
| qSOFA (≥2) | 0.62 | |
| qSOFA + cv-ALL | 0.66 | <0.001 |
| MEWS (≥5) | 0.66 | |
| MEWS + cv-ALL | 0.70 | <0.001 |

cv-ALL, Coefficient of Variance Axial Light Loss; LRT, Likelihood Ratio Test; qSOFA, quick Sequential Organ Failure Assessment; OR, odds ratio; CI, confidence interval.

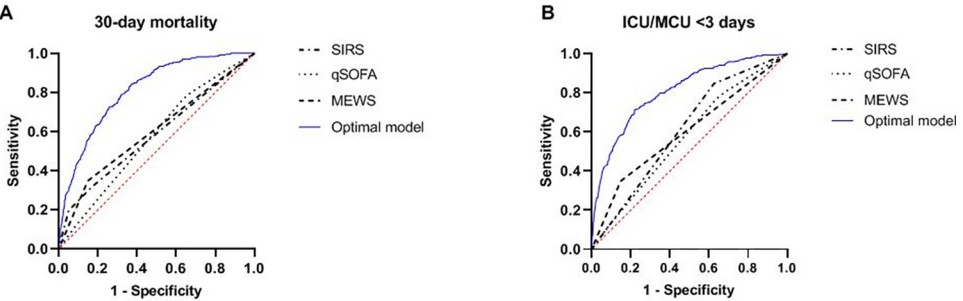

**Fig 2.** ROC curves of the prediction of 30-day mortality (A) and ICU/MCU admission within 3 days after ED presentation (B). The optimal model consisted of the following parameters: cv-ALL of monocytes, age, sex, CCI, respiratory rate, systolic blood pressure, Glasgow Coma Scale, heart rate, leukocyte count, and body temperature. The AUC of the optimal model for prediction of 30-day mortality was 0.81 and for ICU/MCU admission <3 days 0.81 as well.

## 4. Discussion

This is the first study that investigated cv-ALL of monocytes as a biomarker to predict adverse clinical outcomes in patients suspected of an infection at the ED. Our results show that cv-ALL of monocytes could be a usable predictor for both 30-day mortality and MCU/ICU admission for patients that present at the ED and are suspected of an infection. Moreover, cv-ALL of monocytes has additional value to predict mortality and MCU/ICU requirements to the commonly used clinical prediction scores.

Recently, there have been numerous publications on MDW in the context of sepsis [22–27]. However, all previous MDW studies investigated the diagnostic value of MDW for sepsis, rather than its prognostic value. Therefore, it is hard to compare these diagnostic MDW studies with our prognostic study on the cv-ALL of monocytes. A careful comparison can be made, since patients with the diagnosis sepsis are known to have higher adverse outcome rates than patients with less severe conditions [3, 33, 34]. In previous studies, MDW was found able to distinguish SIRS from sepsis-2 [22, 23] as well as to diagnose sepsis based on the sepsis-3 definition with AUCs ranging from 0.73–0.87 [24–26]. In line with this, we found that high values for cv-ALL of monocytes correlate with an increasing risk for adverse clinical outcomes. Additionally, MDW elevation was correlated with infection severity [25, 35] and low values of MDW had strong negative predictive values in the context of sepsis (87–97%, [25–27]). This is similar to our study, which shows that clinically sicker patients have higher cv-ALL of monocytes values.

There are several reasons why cv-ALL of monocytes should be preferred above MDW. Cv-ALL of monocytes is readily available and easily accessible as it can be extracted from a routine hematological analyzer. Consequently, we did not have to perform an extra lab test or buy an extra machine. This implicates major clinical advantages compared to measuring MDW: cv-ALL of monocytes does not require technical knowledge or laboratory space, and is cheaper. Since the essence of the test is so similar to the measurement of MDW and since our results point in the same direction to previous results, it is likely that cv-ALL of monocytes can replace MDW.

Our study has some limitations. First, we imputed missing data of some of the variables, up to 25% for respiratory rate. This may have influenced the performance of our models. We acknowledge that these data may not be missing completely at random. Yet, because severely ill patients have more complete EHR records [36], it is likely that in less severely ill patients more imputation was required. Therefore, undocumented abnormal respiratory rates in less

severely ill patients might be imputed within the normal range. Because of this, we only believe that imputation could have led to underestimation of our results. Second, the study was performed at the UMCU, a large tertiary hospital that is known for its relatively large population of immunocompromised patients. In a subanalysis, cv-ALL of monocytes differed significantly between immunocompromised and non-immunocompromised patients, indicating that immunosuppression affects cv-ALL of monocyte values. Nonetheless, even in this academic population cv-ALL of monocytes appears to be a predictor for adverse clinical outcomes. At last, we show that multivariate models can achieve good AUCs to predict outcome. However, both calibration plots show overestimation in high risk patients, which might be due to the low number of patients with high prediction scores. Therefore, except for ruling out, these models would not be suitable for clinical implementation yet.

There are a few specific strengths to this study. First, current sepsis guidelines advise using qSOFA in the ED setting to predict clinical outcome as opposed to using it as a diagnostic tool [1, 13]. Adding this to the absence of a gold standard to diagnose sepsis at the ED, we decided upon a prognostic study with well-defined outcome measurements rather than a diagnostic design. Moreover, the SPACE-cohort has a well-defined and clinically relevant patient domain, namely all patients at the ED that are suspected of an infection. The heterogeneity resulting from this cohort might explain the relatively low performances of the clinical scores, when compared to other literature [11, 14].

## 5. Conclusion

This study shows that cv-ALL of monocytes is a valuable predictor for 30-day mortality and MCU/ICU requirement <3 days after ED visit in patients suspected of infection at the ED. The clinical performance is likely to be equal to MDW. Nevertheless, cv-ALL of monocytes has multiple practical advantages compared to MDW, making cv-ALL of monocytes more preferable.

## Supporting information

**S1 Fig. Cv-ALL of monocytes is associated with disease severity.** SIRS (A), qSOFA (B), and MEWS (C) score and height of cv-ALL of monocytes is shown. A one-way ANOVA was performed to test group differences. Significance testing was done by Tukey's test. **p < 0.01, ***p < 0.001.
(TIF)

**S2 Fig.** High cv-ALL of monocytes percentage for SIRS (A), qSOFA (B), and MEWS (C) scores. High cv-ALL of monocytes was defined as the cut-off value for our clinical model to predict 30-day mortality (0.085). P-values were calculated with a X-square test.
(TIF)

**S3 Fig. Cv-ALL of monocytes in non-immunocompromised (-) and immunocompromised (+) patients.** Cv-ALL of monocytes differed significantly between these two groups. P-value was calculated by a Mann-Whitney U test.
(TIF)

**S4 Fig.** Optimal model calibration plots for 30-day mortality (A) and ICU/MCU admission <3 days (B). The dashed line shows the calibration plot for the optimal models. The model for 30-day mortality has $R^2$ of 0.209 and Brier score of 0.054, while the model for ICU/MCU admission <3 days shows $R^2$ of 0.220 and Brier score of 0.070.
(TIF)

## Author Contributions

**Conceptualization:** Titus A. P. de Hond, Imo E. Hoefer, Saskia Haitjema, Karin A. H. Kaasjager.

**Data curation:** Titus A. P. de Hond, Wout J. Hamelink, Mark C. H. de Groot.

**Formal analysis:** Titus A. P. de Hond, Wout J. Hamelink.

**Investigation:** Titus A. P. de Hond, Mark C. H. de Groot, Saskia Haitjema.

**Methodology:** Titus A. P. de Hond.

**Project administration:** Titus A. P. de Hond.

**Supervision:** Jan Jelrik Oosterheert, Karin A. H. Kaasjager.

**Writing – original draft:** Titus A. P. de Hond, Wout J. Hamelink.

**Writing – review & editing:** Titus A. P. de Hond, Imo E. Hoefer, Jan Jelrik Oosterheert, Saskia Haitjema, Karin A. H. Kaasjager.

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
