## [Decision Letter · Decision Letter 0]

13 Apr 2022

PONE-D-21-39689Axial light loss of monocytes as a readily available prognostic biomarker in patients with suspected infection at the emergency departmentPLOS ONE

Dear Dr. de Hond,

Thank you for submitting your manuscript to PLOS ONE. After careful consideration, we feel that it has merit but does not fully meet PLOS ONE’s publication criteria as it currently stands. Therefore, we invite you to submit a revised version of the manuscript that addresses the points raised during the review process.

 This is an interesting paper that offers promising preliminary results in an important subject area. In addition to the reviewer comments, please consider the following in your revisions:-can you explain what "semi-automated" data collection is? how much was done by computer vs hand and whas there any manual checking of data pulled automatically?-please include a detailed explanation of missing data-please describe the multiple imputation process in more detail-please include details about model calibration-the discussion is well-written overall and makes some very good points. however, i think that the tone is a bit too conclusive in that this is a single retrospective study from a single center. In addition, automated variable selection techniques are notorious for creating over-fit models--while i think this is a reasonable first step, i do not think these data can be considered definitive. furthermore,  prospective validation is definitely needed before this should be used in routine clinical practice. i would recommend that the discussion be modified slightly to reflect this fact.

We look forward to receiving your revised manuscript.

Kind regards,

Robert Ehrman, MD, MS

Academic Editor

PLOS ONE

Journal Requirements:

Reviewers' comments:

Reviewer's Responses to Questions

**Comments to the Author**

1. Is the manuscript technically sound, and do the data support the conclusions?

Reviewer #1: Yes

2. Has the statistical analysis been performed appropriately and rigorously? 

Reviewer #1: Yes

3. Have the authors made all data underlying the findings in their manuscript fully available?

Reviewer #1: Yes

4. Is the manuscript presented in an intelligible fashion and written in standard English?

Reviewer #1: Yes

5. Review Comments to the Author

Reviewer #1: PONE-D-21-39689

It is a little unclear whether or not all the patients in the study timeframe were included if they were >=18 and had a suspected infection or whether or not it required an abnormal SIRS or qSOFA score.

This was a post-hoc analysis of a prospectively collected database and blood samples. The cv-ALL contributed to improving the performance of know disease severity scores. That said the baseline characteristics of the survivors vs. non-survivors were not balanced. The OR of qSOFA and MEWS was much more robust for the selected outcomes that cv-ALL alone.

Was lactate included in the analysis? cv-ALL is independent of overall WBC? It would suggest yes – given that they are both in the final model, however one would think there is significant interaction b/w these variables.

What is the rationale for a higher CCI being less likely to be admitted to the MCU/ICU – is it related to ICU-restrictions of these patients?

I would have liked to see some basic data concerning process measures such as time to antibiotics which is strongly correlated with outcome in the sicker sepsis/septic shock patients.

This is an interesting response to the MDW theme and offers an alternative that may be more accessible and affordable to institutions interested in pursing such options.

6. PLOS authors have the option to publish the peer review history of their article (what does this mean?). If published, this will include your full peer review and any attached files.

Reviewer #1: No

---

## [Author Response · Author response to Decision Letter 0]

30 May 2022

All raised questions by the academic editor and reviewers were answered in the attached rebuttal letter entitled "response to reviewers".

---

## [Editor Report · Decision Letter 1]

8 Jun 2022

PONE-D-21-39689R1Axial light loss of monocytes as a readily available prognostic biomarker in patients with suspected infection at the emergency departmentPLOS ONE

Dear Dr. de Hond,

Thank you for submitting your manuscript to PLOS ONE. After careful consideration, we feel that it has merit but does not fully meet PLOS ONE’s publication criteria as it currently stands. Therefore, we invite you to submit a revised version of the manuscript that addresses the points raised during the review process.

Thank you for taking the time to revise the manuscript--I think that it is substantially improved. My remaining concern relates to non-reporting of model calibration in the current iteration of the paper (NB: HL test is a goodness of fit measure rather than one of calibration). To me, regardless of how (or even if) one intends prediction models to be used, calibration is an important data point to include--calibration and discrimination (eg, AUC) go hand-in-hand. This is also the recommendation from TRIPOD guidelines (https://www.equator-network.org/reporting-guidelines/tripod-statement/). Please see the reference below for a more detail description of why I (and others) feel calibration is of great import. https://bmcmedicine.biomedcentral.com/articles/10.1186/s12916-019-1466-7 I would also note that sub-optimal model calibration, in and of itself, would in no way preclude publication. if this were the case, it would be an important limitation to address/discuss. i think that this study has value to add to the medical literature and that it is important that results be reported fully and transparently so that researchers who take up this subject matter have as much information as possible when planning and executing further studies.

We look forward to receiving your revised manuscript.

Kind regards,

Robert Ehrman, MD, MS

Academic Editor

PLOS ONE
---

## [Author Response · Author response to Decision Letter 1]

16 Jun 2022

The response is given in the file titled "Response to reviewers".

---

## [Editor Report · Decision Letter 2]

20 Jun 2022

Axial light loss of monocytes as a readily available prognostic biomarker in patients with suspected infection at the emergency department

PONE-D-21-39689R2

Dear Dr. de Hond,

We’re pleased to inform you that your manuscript has been judged scientifically suitable for publication and will be formally accepted for publication once it meets all outstanding technical requirements.

Kind regards,

Robert Ehrman, MD, MS

Academic Editor

PLOS ONE
---

## [Editor Report · Acceptance letter]

1 Jul 2022

PONE-D-21-39689R2 

Axial light loss of monocytes as a readily available prognostic biomarker in patients with suspected infection at the emergency department 

Dear Dr. de Hond:

I'm pleased to inform you that your manuscript has been deemed suitable for publication in PLOS ONE. Congratulations! Your manuscript is now with our production department. 

Kind regards, 

on behalf of

Dr. Robert R Ehrman 

Academic Editor

PLOS ONE